# Resuscitation simulation among people who are likely to witness opioid overdose: Experiences from the SOONER Trial

Jonathan P. Whittall[1], Aaron M. Orkin[1,2,3], Curtis Handford[1], Michelle Klaiman[4], Pamela Leece[5], Mercy Charles[6], Amy Wright[5,7], Suzanne Turner[8], Laurie J. Morrison[9], Carol Strike[10], Douglas M. Campbell[6,11]*

1 Department of Family and Community Medicine, University of Toronto, Toronto, Ontario, Canada, 2 Inner City Health Associates, Toronto, Ontario, Canada, 3 Department of Emergency Medicine, St. Joseph's Health Centre, Unity Health Toronto, Toronto, Ontario, Canada, 4 Department of Emergency Medicine, St. Michael's Hospital, Unity Health, Toronto, Ontario, Canada, 5 Public Health Ontario, Toronto, Ontario, Canada, 6 Applied Health Research Centre (AHRC), Li Ka Shing Knowledge Institute, St. Michael's Hospital, Unity Health, Toronto, Ontario, Canada, 7 Canada SOONER Project Community Advisory Committee, St. Michael's Hospital, Unity Health, Toronto, Ontario, Canada, 8 Department of Family Medicine, McMaster University, Hamilton, Ontario, Canada, 9 Rescu, Li Ka Shing Knowledge Institute, St. Michael's Hospital, Toronto, Ontario, Canada, 10 Dalla Lana School of Public Health, University of Toronto, Toronto, Ontario, Canada, 11 Department of Paediatrics, University of Toronto, Toronto, Ontario, Canada

* Douglas.Campbell@unityhealth.to

**Data Availability Statement:** Data cannot be shared publicly as indicated in the protocol and consent, following all privacy legislation in Ontario. Data are available from the secure and encrypted

## Abstract

The opioid crisis is a growing public health emergency and increasing resources are being directed towards overdose education. Simulation has emerged as a novel strategy for training overdose response, yet little is known about training non-clinicians in bystander resuscitation. Understanding the perspectives of individuals who are likely to experience or witness opioid overdose is critical to ensure that emergency response is effective. The Surviving Opioid Overdose with Naloxone Education and Resuscitation (SOONER) study evaluates the effectiveness of a novel naloxone education and distribution tool among people who are non-clinicians and likely to witness opioid overdose. Participants' resuscitation skills are evaluated using a realistic overdose simulation as the primary outcome of the trial. The purpose of our study is to describe the experience of participants with the simulation process in the SOONER study. We employed a semi-structured debriefing interview and a follow up qualitative interview to understand the experience of participants with simulation. A qualitative content analysis was performed using data from 21 participants who participated in the SOONER study. Our qualitative analysis identified 5 themes and 17 subthemes which described the experience of participants within the simulation process. These themes included realism, valuing practical experience, improving self-efficacy, gaining new perspective and bidirectional learning. Our analysis found that simulation was a positive and empowering experience for participants in the SOONER trial, most of whom are marginalized in society. Our study supports the notion that expanding simulation-based education to non-clinicians may offer an acceptable and effective way of supplementing current opioid overdose education strategies. Increasing the accessibility of simulation-based education may

server and data management system. Institutional Data Access / Ethics Committee (contact via request) for researchers who meet the criteria for access to confidential data. Interested and qualified researchers may submit data access requests to Unity Health Toronto at researchcontracts@smh.ca.

**Funding:** The funders had no role in study design, data collection and analysis, decision to publish, or preparation of the manuscript.

**Competing interests:** The authors have declared that no competing interests exist.

represent a paradigm shift whereby simulation is transformed from a primarily academic practice into a patient-based community resource.

## Introduction

Simulation engages learners in a contextualized educational environment, in which clinical scenarios are reproduced as closely as possible to improve performance. Simulation has evolved into a major tool within healthcare provider education and is the reference standard for teaching and evaluating resuscitation skills [1–3]. Given its well-documented success as an educational tool for clinicians, simulation is now more mainstream, and more non-clinicians are increasingly experiencing it for health-related purposes [4, 5].

The opioid crisis is an ongoing public health emergency which is receiving increasing attention and public funding across North America [6, 7]. Over the past decade, simulation has emerged as a novel strategy for training overdose response and for evaluating the effectiveness of overdose education initiatives [8]. Non-clinicians are increasingly being introduced to simulation in the context of simulated opioid overdose [9–12]. While the experience of healthcare trainees with simulation has been well characterized, little is known about how simulation is experienced by individuals who are not trained in health care and who are likely to experience or witness an overdose.

People who use high dose opioids or opioids from the toxic street market are more likely to experience or witness overdose and it is imperative that these individuals be included in the development of opioid overdose education initiatives [8]. Although conducting trials with this population involves logistical, sociocultural and bioethical challenges, stigmatization within our healthcare system may contribute to this population's limited exposure to simulation to date [13]. It has been well recognized that individuals who use opioids from the toxic street market have higher rates of trauma than the general population and are often marginalized in society [14, 15]. Many individuals who use drugs also have experience with opioid overdose, an extremely stressful event which has been associated with high rates of unrecognized psychological trauma [16]. Simulation can be an emotionally activating experience and it is not known whether exposing this population to a realistic overdose scenario could be psychologically triggering or harmful [17]. Understanding the experience of individuals who are likely to witness or experience overdose is important to ensure that simulation is well implemented given its increasing use with this population.

The Surviving Opioid Overdose with Naloxone Education and Resuscitation (SOONER) study is a Toronto-based initiative in which individuals who were identified as being at high risk of experiencing or witnessing an opioid overdose were immersed into a high-fidelity simulated overdose. The primary objective of the SOONER study was to evaluate whether a novel point-of-care naloxone distribution and education strategy could lead to improved resuscitation outcomes compared to the current standard of care which involves a referral to a local pharmacy or naloxone distribution site [18, 19]. A randomized controlled trial was designed in which participants were allocated to either receive the point-of-care naloxone distribution strategy or the community referral strategy. Within two weeks of randomization, they were invited back to participate in a contextualized simulation in which their skills as a responder to opioid overdose were evaluated. The ability of participants to successfully resuscitate the mannikin (administer naloxone, perform CPR, etc.) was evaluated and their performance during simulation was used as the primary outcome measure for the SOONER study [18].

To our knowledge, SOONER is the first study to immerse a population who is specifically selected for their high likelihood of experiencing or witnessing an overdose in simulation. A unique opportunity is therefore created to try to understand what it is like for this population to participate in a high-fidelity simulated overdose. The purpose of this study is to describe the experience of participants with the simulation process in the SOONER study. The objective performance of participants during the simulated overdose represents the outcome measure of the SOONER study and will not be described in this paper. Rather, we seek to understand the perspectives and the subjective experience of participants with the simulation process.

## Methods

### Study design

This study employed a semi-structured debriefing interview and a follow up qualitative interview to understand the experience of participants in the SOONER trial [18]. It should be noted that the scope of this study pertained specifically to the experience of participants with the simulation process and that discussions that arose around broader themes such as addiction, stigma and rehabilitation are not described in this paper.

### Setting and sample

Participants consisted of individuals over the age of 16 identified at high risk of opioid overdose or the friends and family of people at risk of witnessing an overdose. Detailed inclusion and exclusion criteria can be found in the published protocol for the SOONER study [18]. Participants were recruited from multiple hospital-based services at St. Michael's Hospital, an urban, academic, tertiary care, inner-city hospital in Toronto, Ontario, Canada. Recruitment occurred in the Emergency Department, the outpatient Addiction Medicine Clinic, and the inpatient Addiction Medicine Service. Outpatients were also recruited from the St. Michael's Hospital Academic Family Health Team and the Inner-City Family Health Team, which serves over 50,000 patients. The recruitment and retention strategies for the study were developed in collaboration with community representatives and drawing on available literature about retaining underserved populations in research studies [18]. A candidate-driven recruitment strategy was used and adapted for implementation in each of the recruitment settings (ED, family practice, addictions medicine). Participant retention was supported with a variety of strategies including flexible scheduling, reminders, and incentives. The details of the recruitment and retention strategy are described elsewhere [18, 19]. The simulation and interviews took place at the Allan Waters Family Simulation Centre at St. Michael's Hospital, Unity Health Toronto.

Demographic data for the 21 individuals who completed the SOONER study are presented in Table 1. The median age of participants was 43 and most participants identified as male (67%) and White (70%). Thirty three percent of participants reported living on the streets or in a shelter (33.3%) and over half of participants reported regular non-prescription opioid use (57%). Most participants endorsed previous experience with overdose having either witnessed (56%) or experienced an overdose themselves (33%). Less than half of the participants in our study identified having received formal CPR or opioid overdose education training (42.8%).

### The SOONER simulation protocol

The simulation protocol used in the SOONER study was composed of 3 structured components: a briefing session, the simulated overdose scenario, and a debriefing session [18].

**Table 1. Participant demographics.**

| Category | Overall (N = 21) |
|---|---|
| **Age (median [IQR])** | 43.00 [35–54] |
| **Self-reported gender** no. (%) | |
| Male | 14 (66.7) |
| Female/other | 7 (33.3) |
| **Born in Canada** no. (%) | 16 (76.2) |
| **Self-reported ethnicity/race** no. (%) | |
| White | |
| Other | |
| **Housing** no. (%) | |
| Renting | 8 (38.1) |
| Shelter/Homeless/Group home | 7 (33.3) |
| Other | 6 (28.6) |
| **Education** no. (%) | |
| Elementary | 6 (28.6) |
| High school | 5 (23.8) |
| College/university | 10 (47.6) |
| **Current opioid use** | |
| Rx opioids (hydromorphone, morphine, etc.) | 6 (28.6) |
| Non-Rx opioids (fentanyl, heroin, etc.) | 12 (57.1) |
| OAT (methadone, buprenorphine) | 7 (33.3) |
| **Experience with overdose** | |
| Has witnessed an opioid overdose | 10 (55.6) |
| Has personally required emergency care for an opioid overdose | 7 (33.3) |
| **Previous first aid, CPR or OEND training experience** no. (%) | 9 (42.9) |

The briefing was a standardized video that oriented participants to the room and simulation scenario. It instructed them to respond as they would if the simulation were real and aimed to create a safe learning environment. A video briefing was chosen to ensure that study participant experience was as consistent and standardized as possible throughout the simulation (protocol in S1 Text).

The simulation protocol was based on the 2015 American Heart Association bystander resuscitation recommendations and adapted based on similar studies to mimic a realistic overdose situation. Design and testing of the scenario was done with community members who had lived experience with opioid overdose. The simulation began as the participant walked into a staged bedroom to find a manikin propped up next to a bed with an empty syringe on the ground (Fig 1). The Laerdal SimMan® ALS manikin, an advanced patient simulator with capabilities that include vital signs such as a palpable pulse and breathing was used for the scenario. A phone was placed in the room to allow participants to simulate a 911 call, which was answered by the research coordinators. The simulation lasted between 5–10 minutes based on participant response to each event and ended with the "arrival of 911" which was signaled by a siren and the entrance of the research coordinators into the room. Importantly, the simulation was designed to mimic a very near-fatal overdose. Without prompt intervention, the simulated patient's physiological status would proceed to cardiac arrest and death. To achieve a successful resuscitation, patients were required to perform what they had been taught during their educational intervention. At a minimum, participants were required to administer intranasal naloxone and perform chest compressions on the manikin but their ability to recognize the

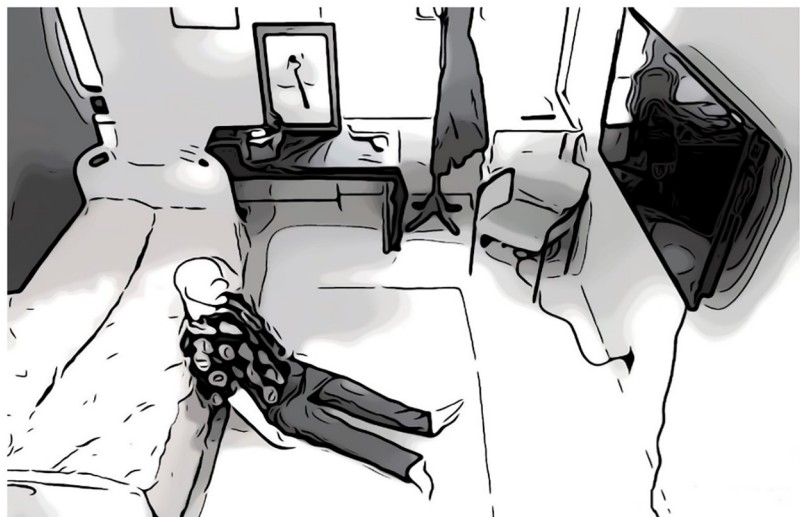

**Fig 1. Schematic of the simulation room.**

emergency, position the manikin, activate Emergency Medical Services (EMS), and organize their approach was also evaluated. The overdose simulation was recorded, and the ability of participants to navigate through the resuscitation algorithm and complete each of its steps was evaluated independently by two clinicians blinded to allocation.

A debriefing session was conducted by the research coordinator immediately following the simulation. Since no tool has been validated to lead a simulation debriefing with non-clinicians who have witnessed opioid overdoes, the Promoting Excellence and Reflective Learning in Simulation (PEARLS) framework was adapted to guide the debriefing. Developed by Eppich & Cheng in 2015 for use with healthcare learners, the PEARLS framework combines standard scripted language with an adaptable framework and designed to promote reflection of learning and performance [20]. Through facilitated discussion, participants were asked to self-assess their performance and reflect on their experience with simulation. During this debriefing session research coordinators affirmed positive behaviors, provided teaching, and offered an opportunity for participants to ask questions and clarify misunderstandings.

## Data collection

Data from two semi-structured interviews were used for the qualitative analysis. The first source of data was the debriefing which occurred immediately following the simulated overdose and was guided by the Promoting Excellent and Reflecting Learning in Simulation (PEARLS) framework. Participants were informed that this session was to help consolidate learning and understand their experience with the simulation process. The video recorded debriefing was led by the evaluating clinicians and lasted approximately 5–10 minutes (protocol in S2 Text). The second source was a more in-depth qualitative interview which occurred following the debriefing and was administered by researchers involved with the SOONER trial (protocol in S3 Text). During this second interview, participants were asked about their overall experience with the research project, including the simulation session, and probing questions were used by researchers to expand on ideas brought up by participants. An interview guide was created to facilitate this interview though its structured was flexible to encourage participant-driven discussion. The qualitative interview was audio recorded and data from the interviews were coded and stored in a secure study database. Data pertaining specifically to the

experience of participants with the simulation process was used for the analysis in this study. Quotes from both interviews are used below as exemplars and annotated accordingly in the results (I = interview, D = debriefing).

## Content analysis

Braun and Clarke's 6 step framework was used to guide a qualitative content analysis [21]. For this analysis, we focused specifically on coding of the interview content pertaining to the experience of participants with the simulation process. Coding was manually performed by the primary author of this paper who did not have any role in the interviews or design of the SOONER trial. In accordance with Braun and Clarke's analytic strategy, we first familiarized ourselves with the data by listening to and then transcribing verbatim the recorded patient interviews. Data pertaining to the simulation experience was identified and subdivided into fragments of meaning which were assigned in-vivo codes. Codes were assessed in relation to one another, and an inductive process was used to identify emerging themes. Deductive reasoning was then used to identify similarities and differences between the themes and an iterative process was used to consolidate and subdivide themes. Themes were described in relation to one another and defined in the context of the data set as a whole.

## Ethics

This study received ethics approval and ongoing oversight by the Research and Ethics Board of the Unity Health Toronto Research Ethics Board (REB), Toronto Public Health Research Ethics Board, and the University of Toronto Research Ethics Board (REB). A systematic review on strategies to improve informed consent processes found that having a one-to-one discussion was the most effective way of improving participant understanding of the research process. Verbal consent was therefore obtained from participants in a setting that afforded ample time for discussion and questions. Study coordinators then documented that they obtained verbal consent on the paper consent.

## Results

Thirty participants were recruited to participate in the SOONER study. Nine participants were lost to follow up and twenty-one (21) participants completed the simulation protocol. Patients were recruited from the addiction medicine service, emergency department and family medicine clinics at St. Michael's Hospital, Unity Health Toronto. The first sixteen participants who completed the study took part in a second qualitative interview, a sample size which was defined by thematic saturation.

Data pertaining to the simulation experience were abstracted from 16 qualitative interviews and 21 debriefing sessions. A total of 284 transcript excerpts were identified that pertained to the experiences of participants with simulation. Qualitative content analysis identified 5 major themes and 17 subthemes and thematic saturation was achieved (Table 2).

## Theme #1: Realism

"I've been involved with it before, and it felt the same" (D-P9). Physical fidelity, emotional stress and memory reactivation were identified as key contributors to the realism of the simulation experience. Physical fidelity refers to the ability of the manikin and the staged environment to reproduce real life [22]. The manikin's ability to simulate a human was a common discussion point during the debriefing: "The manikin is a really good idea because it feels like a real human" (D-P17). In addition to the ability to simulate breathing and a pulse wave, the

**Table 2. Results from the qualitative content analysis.**

| | No. of occurrences | Sample Quote |
|---|---|---|
| **1. REALISM** | | |
| 1A. Physical fidelity | 22 | [The manikin] feels and weighs the same as a real human. (D-21) |
| 1B. Emotional activation | 23 | Now I can breathe, that was wild. (D-P8) |
| 1C. Memory reactivation | 23 | I just saved my friend, so I was picturing her during the simulation. (I-P8) |
| **2. VALUING PRACTICAL EXPERIENCE** | | |
| 2A. Hands-on learning | 10 | You can't just read the steps, you have to practice it. (I-P6) |
| 2B. Importance of practice/repetition | 14 | Its mostly practice that you need to get comfortable with this. (D-P3) |
| 2C. Identifying knowledge gaps | 20 | I didn't know that I needed to do continuous CPR. (D-P5) |
| 2D. Developing process skills | 8 | It's hard to talk on the phone and go back and forth. I have difficulties multi-tasking. (D-P3) |
| **3. INCREASING SELF-EFFICACY** | | |
| 3A. Improving confidence | 19 | Just having this hands-on experience definitely makes you feel a lot more confident. (I-P16) |
| 3B. Positive user experience | 14 | [the simulation] was perfect—it was more than what I thought it would be. (I-P1) |
| 3C. Altruism | 23 | I joined the study to learn how to save someone's life. (I-P9) |
| 3D Valuing the training/knowledge | 35 | I feel lucky because I am getting paid to learn. I would have done this just for the knowledge. (D-P1) |
| **4. GAINING NEW PERSPECTIVE** | | |
| 4A. Simulation as a new experience | 11 | It was my first-time taking part in an experience like this. (D-P14) |
| 4B. Self-reflection | 7 | I'm still here, you get to a point where you wonder why. (D-P17) |
| 4C. Developing a different perspective | 6 | I have overdosed several times, but I never realized what it is like to be on the other side. (D-P3) |
| **5. BIDIRECTIONAL LEARNING** | | |
| 5A Identifying barriers to resuscitation | 18 | The good Samaritan act doesn't help, the police still arrest you. (D-P8) |
| 5B. Sharing lived experience | 21 | There are ODs [at the shelter] every single day, so its common to find someone unresponsive there. (I-P7) |
| 5C. Desire to share the experience | 10 | Things like this should be out there more. I haven't seen anything like this. (D-P4) |

manikin's weight was frequently discussed, and many participants felt that "[the weight] made it feel so real" (D-P12). While the ability of a manikin to replicate a real human is limited, individuals are more willing to accept lapses in physical fidelity if conceptual fidelity is maintained [23]. This was explained by participants: "I've been involved with [overdoses] almost double-digit times and it felt the same. It's always been someone passed out with a needle next to them, same kind of situation" (I-P7).

Emotional stress refers to the emotional activation and stress response experienced by participants during the overdose simulation. Most participants reported feeling "anxious", "stressed", or "nervous" during the simulation. A minority of participants acknowledged feeling "overwhelmed" (D-P15) or "bewildered" (D-P17) and one individual stated that "[he didn't] know if [he] was prepared for it" (D-P3). Despite these reactions, participants valued the way that simulation made them feel: "It got me into it more than I thought. Which is a good thing." (D-P1). "It gave me a good idea of how it could be in a situation like that" (I-P1) said another.

While participating in an observed simulation is inherently stressful, one participant believed that "[the experience] may be different for a person who is in active recovery [from a substance use disorder] compared to someone who is completely sober (I-P6)". Memory reactivation was a common discussion point and may account for this difference: "When I walked into the room and saw everything, and I had a déjà vu moment" (I-P14). As another participant recounts:

I just went through [an overdose] the other day. Before she injected, I told her to put in 2 rigs because I knew it was carfentanyl, but she said no I am fine. She did it and the next

thing I knew she seized right up and went totally stiff and blue. That's what just came to me as I was doing that.

(D-P5)

By reactivating pre-existing memories, experiences with overdose may increase an individual's emotional reaction to the scenario further enhancing simulation's realism. Overdose simulation may therefore be experienced differently and feel more real to someone who has had previous similar experiences compared to someone who has never been involved with an opioid overdose.

## Theme 2: Valuing practical experience

Participants valued the opportunity to practice resuscitation skills during simulation. Although study participants had been given an explanation on how to use naloxone and perform chest compressions, many had never actually practiced these skills: "I've never even sprayed [naloxone] before. That was the first time I've ever done it" (D-P3). Others believed that resuscitation skills could not be acquired without hands on experience: "you can read about it but until you put your hands to work and do it, it's the only time you're actually going to get it" (I-P6). Many participants were surprised by the ease with which naloxone could be administered with many acknowledging that they had developed misperceptions and cognitive barriers when it came to using naloxone: "administering the nasal spray was easy, if it was a needle, I wouldn't have felt comfortable" (D-P15). "I will carry the spray permanently now" (I-P1) vowed one participant following simulation.

For individuals who reported previous experience responding to opioid overdoses, the importance of practice and repetition emerged as an important theme: "How can anyone say that this practice session is not worthwhile. It puts it at the front of your mind" (D-P4). One participant used the analogy of riding a bike to describe learning overdose resuscitation skills.

The first time you ride a bike you have to concentrate really hard and you have to try it over and over again. Responding to an overdose is the same although it's unfortunate because someone's life is always at stake. You won't be able to get it if you don't practice it.

(I-P6)

Participants also valued the opportunity to practice higher order cognitive skills during simulation. Multitasking and prioritizing steps in the resuscitation algorithm were frequently discussed as being challenging: "I felt like the multitasking was tough especially not having been in this situation ever." As another participant reflects: "It was what comes first that was my biggest problem. I see that he is not breathing and that his eyes are small, but I wasn't sure where to start" (D-P20).

The practical nature of simulation also allowed many participants to discover previously identified knowledge gap. Given the unique ability of the manikin to provide objective feedback on chest compressions, the technique of performing chest compressions was a common discussion point:

It's not my first-time doing (chest compressions) but I now realize the depth was wrong and I should have been going harder. I walked in here thinking that I knew what I was doing but was humbled. I learned something so I think that this is really important

(I-P7).

Following the simulation experience, many participants expressed a desire for more wide-spread opportunities to practice resuscitation skills: "Things like this should be out there more" (D-P4).

## Theme 3: Increasing self-efficacy

"I felt moderately comfortable before and absolutely comfortable now" (I-P3). By allowing participants to gain experience with both discrete and process skills, simulation allowed participants to develop confidence in their overdose resuscitation skill: "Now I would definitely know what to do if it was a real human" (D-P21).

In accordance with self-determination theory, self-efficacy can be enhanced if individuals are given autonomy in an environment in which they feel respected and well supported [24], something that participants agreed with: "it was comfortable, everything was comfortable, and it was professional" (I-P6). Simulation was described as an overwhelmingly positive experience by participants: "It was perfect actually, it was more than I thought it would be" (I-P1). "I can't think about anything negative about this simulation" (D-P1) added another.

Most participants readily identified strengths of their performance and positive self-talk was common during the semi-structured interviews. "I thought I did all of it really well" (I-D17) said one participant. Many participants also told stories of responding to previous overdoses and described past success during the debriefing: "I've done things since being in Toronto that people are actually amazed with" (I-P10). "I have a lot of street smarts" (I-P6) said another.

For many participants, overdose response skills were a clear source of pride and a driver of self-efficacy. A deep sense of altruism appeared to drive study participants: "I joined the study because I wanted to do something to help out others" (I-P9). "It's just a humanitarian thing to do" (I-P7) said another. Christian faith, civil duty, and empathy were also cited as reasons that participants chose to participate in the study. Many participants also felt that their background with drug use and addiction made the training more relevant: "It's helpful to learn this type of stuff because I use opioids, and I am surrounded by people who use opioids" (I-P9). As another participant explains:

> Now that I am in recovery, I get to see a lot of people that are suffering from this epidemic—you know people dying around me and it's sad to see. If I see someone in pain or suffering, I am willing to help if I can because I have been around drugs.
>
> (I-P9)

By aligning with personal values and allowing participants to develop overdose response skills, simulation offered an experience that was identity affirming for many participants. Simulation may therefore increase self-efficacy by allowing participants to develop confidence in skills that they value.

## Theme 4: Gaining new perspective

For many participants, the simulation process stimulated powerful introspective reflection. Simulation was described as "eye-opening" (I-P1), a "wake up call" (D-P3) and an "insightful experience" (D-P6) by participants. Many participants believed that their past experiences influenced the way that they experienced simulation.

For many participants who endorsed taking drugs or a history of overdose, overdose simulation was deeply personal. Following simulation, participants were eager to share stories, and many discussed their own struggles with addiction: "I'm still here, you get to a point where

you wonder why. I am probably 30 years past my due date. (D-P17)". For others, simulation offered a new perspective: "I have overdosed several times, but I never realized what it is like to be on the other side. I usually just wake up and wave everyone away" (D-P3). By offering participants an opportunity to take on the resuscitative role, simulation provided a different vantage point and a new perspective on overdose.

For individuals without any previous overdose experience, the experience was new: "I've never experienced anything like this". Simulation also allowed participants to gain an appreciation for what responding to an overdose would be like: "[Simulation] made me realize how stressful it would be in a real situation" (D-P3). Participants particularly valued the insight that simulation allowed them to gain into their own reactions and emotions: "It's going to be a different experience for everyone. The steps may be the same, but everyone is going to respond differently" (I-P6). As one participant reflects: "I froze a bit when I first came in—those extra couple seconds can mean a lot. And even after I gave him naloxone I stood back." (D-P1). Participants appreciated the insight that simulation allowed them to gain: "This is a great experience in case it happens to me (D-P6).

## Theme 5: Bidirectional learning

The semi-structured debriefing which was included in SOONER's design was intended as an opportunity for researchers to engage participants around the simulation process and offer overdose resuscitation teaching. While this goal was accomplished, participants were also eager to share insight from their lived experience, and the debriefing became a forum for bidirectional learning. The decision to activate 911 became an interesting point of discussion during the debriefing session:

> It's better that the bystander stays but in real life it never happens. What happens is that people administer naloxone, they call 911 and then they take off because the police are going to come. If the police come they start arresting people—even if it isn't my drugs they will arrest me if I am in the presence of it.
>
> (D-P8)

By allowing space for participants to share their insight, the debriefing allowed participants offer feedback which was valued by the evaluating clinicians. In addition to the fear of police, the fear of hurting another person, making a mistake, or of getting a friend in trouble were identified as limitations which could prevent the successful implementation of the resuscitation plan: "I wouldn't want them to be exposed, I know their life is being threatened but it's still a tricky situation" (D-P11).

Many participants were eager to offer their support for simulation education: "stuff like this is not available. It is not advertised" (D-P13). "Things like this should be out there more" (D-P4) said another. Real life experience was described as the primary source of overdose resuscitation experience by many: "I haven't seen anything like this. The only time I've seen it is in real life" (D-P4). Participants were also eager to offer suggestions on how to improve the experience: "In a real overdose situation you have people screaming in the background so if you wanted to make it feel more real you could have people freaking out in the background with no one knowing what to do". Many individuals believed that simulation should be more accessible: "I definitely think that anyone that uses these drugs should get trained on this stuff" (D-P3). Others recommended targeting certain groups: "you guys should teach youth this stuff because they don't know anything about [overdose resuscitation]" (D-P13). "Service like this

should be available outside [shelters and treatment facilities]. I think a lot of people going through treatment would be willing to learn how to save lives" (D-P13).

## Discussion

This study described the simulation protocol used in the SOONER study and characterized the experiences of participants with the simulation process. Overall, the simulation protocol that included a briefing and debriefing session was well received by participants in the SOONER trial. The modified PEARLS framework [20] stimulated meaningful discussion and resulted in the collection of rich qualitative data, supporting its continued use with non-clinicians, offering both participant and research team to learn from each other.

The results from our analysis found that simulation was a positive and constructive experience for participants in the SOONER trial. The **realistic** and **practical** nature of the experience as well as simulation's ability to **increase self-efficacy** and **offer new perspective** was valued by participants. The debriefing session and qualitative interview offered an opportunity for dialogue which facilitated learning and empowered participants to engage in **bidirectional learning** with the evaluating clinicians.

To our knowledge, this is the first study to characterize the experience of individuals who are likely to experience or witness overdose with simulation. Although the simulation process elicited challenging emotions and retriggered past experiences for many, many participants believed that their lived experiences contributed to realism of overdose simulation and increased the relevance of the training. Findings from this study refute the idea that exposing individuals with previous overdose experience to a contextualized simulation is psychologically damaging. Rather, participants described the simulation process as a comfortable and affirming experience, supporting future inclusion of individuals who use drugs or are likely to witness overdose in simulation.

While programs that teach overdose response are increasing, many of our participants felt that opportunities to practice hands-on skills are still lacking. Given the ongoing high rate of opioid overdoses, participants in our study primarily reported acquiring their overdose response skills by responding to real overdoses and learning from the experiences of others. Resuscitation skills were an important driver of self-efficacy and many of our participants regarded themselves as altruistic individuals who took pride in being resources to their communities. Communities of practice is a term proposed by education theorist Etienne Wenger which describes a group that shares a concern or passion for something they do and learn how to do it better as they interact regularly [25]. While the term is most used in academic settings, the accounts from individuals in our study support the notion that communities of practice may exist within communities of individuals who use drugs in the context of overdose response. Simulation-based education may represent an opportunity to support the ongoing efforts of individuals who use opioids to acquire overdose resuscitation skills.

Currently, simulation-based education is primarily reserved for healthcare learners and is not widely available to non-clinicians who witness overdoses [26]. Multiple recent studies have challenged the idea that simulation requires 'high-technology' equipment to be effective [27]. Low-technology simulation, a form of simulation which initially may be perceived as 'less realistic' to the learner, may be just as effective and more generalizable compared to high fidelity simulation [28, 29]. Taking simulation outside of the laboratory setting may therefore allow for increased accessibility and portability without compromising its effectiveness. In-situ simulation (ISS), the process by which simulation is performed within the learner's usual

environment has also emerged as a standard approach to simulation education. Grounded in situational learning theory which states that training is most effective when it reproduces the context of the intended performance, ISS has been shown to improve learning and improve patient outcomes [30, 31]. In the context of overdose response, an ISS approach would align with the wishes of patients in our study, who recommended that simulation be offered in shelters and other settings in which overdoses occur frequently, or where access to users could happen. Increasing the accessibility of simulation-education to bystanders of opioid overdose could represent a paradigm shift in which simulation is transformed from a primarily academic resource into a more community-based resource.

Lastly, the inclusion of individuals who use drugs or are likely to witness overdose in simulation may offer insight capable of guiding public health interventions. In a recently published paper Kneebone et al. argue that the restriction of simulation to clinical insiders has limited the perspectives of patients which are invaluable to clinical practice [26]. Individuals who use drugs often have lived experience with opioid overdose and it is imperative that they be included in the design of overdose education initiatives. Findings from our analysis support the inclusion of a debriefing session in the simulation protocol as a means of engaging participants and eliciting their feedback. By allowing participants space to reflect, the debriefing session empowered participants to share insight which allowed researchers to identify gaps in the proposed resuscitation protocol and potential strategies for improving the simulation experience. Including individuals with lived experience in overdose resuscitation simulation may therefore offer a unique perspective capable of guiding the development of collaborative, patient-centered initiatives.

Several limitations should be discussed in the context of our findings. First, participants were interviewed by clinicians involved in the design and implementation of the simulation process so a social desirability bias may have influenced participants to report positively on their simulation experience. Such a bias could have led to the underreporting of challenges or perceived limitations associated with simulation as an educational tool. Second, a response bias likely existed in which individuals who signed up to participate in the study and came to the simulation center for testing may have had a positive view of simulation or a desire to participate in such an experience (recruitment bias). Lastly, individuals were asked to reflect on their experience immediately following simulation, so findings from this study should be interpreted cautiously when making inferences on the long-term impacts of simulation on this population.

## Conclusion

Simulation was a positive and affirming experience for participants of the SOONER trial. To our knowledge, this is the first study to characterize the experience of individuals who are likely to experience opioid overdose with overdose resuscitation simulation. Findings from our study support the inclusion of this population in trials that use simulation as an outcome measure or teaching methodology to assist in opioid overdose response. Increasing the accessibility of simulation-based education may offer an opportunity to supplement current opioid overdose education initiatives in a way that is acceptable and valued. Finally, SOONER's simulation protocol promoted bidirectional learning and findings from this study support the inclusion of a debriefing session in the design of simulation protocols used with non-clinicians. More research is needed to understand the long-term impacts of overdose resuscitation simulation and to understand how simulation can be optimized to be most effective with this population.

## Supporting information

**S1 Text. Orientation protocol.**
(DOCX)

**S2 Text. Debriefing (PEARLS debriefing guide).**
(DOCX)

**S3 Text. Qualitative interview guide.**
(DOCX)

## Acknowledgments

We would like to acknowledge those who gave a significant amount of time and expertise in incorporating simulation into the SOONER protocol. We would like to thank the SOONER committee including Kate Sellen, Vicky Stergiopoulos, Daniel Werb, Shaun Hopkins, Richard Hunt, Rita Shahin, Kevin Thorpe, Nick Goso, Peter Juni, Janet Parsons, Brenda Bernet, Meg Masse, Leigh Chapman, Brandon Laford, Geoffrey Milos and Kristine Norris. A special acknowledgement to community members with lived experience of opioid overdose who assisted with the design of the simulation scenario. A further acknowledgement should go out to members of the Allan Waters Family Simulation Team at Unity Health Toronto, in particular: Mr. Hentley Small, Ms. Sue Zelko, Ms. Ashley Rosen, and Ms. Nazanin Khodadoust. Finally, a special thank you to the incredible feedback from the participants of the SOONER study. Your lived experience and feedback were valuable to both the research team and the simulation program. Your voice is valued.

## Author Contributions

**Conceptualization:** Jonathan P. Whittall, Aaron M. Orkin, Douglas M. Campbell.

**Data curation:** Mercy Charles.

**Formal analysis:** Jonathan P. Whittall, Douglas M. Campbell.

**Investigation:** Jonathan P. Whittall, Aaron M. Orkin, Douglas M. Campbell.

**Methodology:** Jonathan P. Whittall, Aaron M. Orkin, Douglas M. Campbell.

**Project administration:** Mercy Charles.

**Supervision:** Aaron M. Orkin, Douglas M. Campbell.

**Writing – original draft:** Jonathan P. Whittall, Aaron M. Orkin, Douglas M. Campbell.

**Writing – review & editing:** Jonathan P. Whittall, Aaron M. Orkin, Curtis Handford, Michelle Klaiman, Pamela Leece, Mercy Charles, Amy Wright, Suzanne Turner, Laurie J. Morrison, Carol Strike, Douglas M. Campbell.

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
