## [Decision Letter · Decision Letter 0]

4 Apr 2022

PONE-D-21-30823Resuscitation simulation among people who are likely to witness opioid overdose:

Experiences from the SOONER TrialPLOS ONE

Dear Dr. Whittall,

Thank you for submitting your manuscript to PLOS ONE. After careful consideration, we feel that it has merit but does not fully meet PLOS ONE’s publication criteria as it currently stands. Therefore, we invite you to submit a revised version of the manuscript that addresses the points raised during the review process.

We look forward to receiving your revised manuscript.

Kind regards,

César Leal-Costa, Ph. D

Academic Editor

PLOS ONE

https://journals.plos.org/plosone/s/file?id=ba62/PLOSOne_formatting_sample_title_authors_affiliations.pdf".

2. Please note that according to our submission guidelines (http://journals.plos.org/plosone/s/submission-guidelines), outmoded terms and potentially stigmatizing labels should be changed to more current, acceptable terminology. To this effect,  “Caucasian” should be changed to “white” or “of [Western] European descent” (as appropriate).

*Please include additional information regarding the interview guide used in the study and ensure that you have provided sufficient details that others could replicate the analyses. Please also  include a copy as Supporting Information.

Reviewers' comments:

Reviewer's Responses to Questions

**Comments to the Author**

1. Is the manuscript technically sound, and do the data support the conclusions?

Reviewer #1: Partly

Reviewer #2: Yes

2. Has the statistical analysis been performed appropriately and rigorously? 

Reviewer #1: N/A

Reviewer #2: Yes

3. Have the authors made all data underlying the findings in their manuscript fully available?

Reviewer #1: No

Reviewer #2: Yes

4. Is the manuscript presented in an intelligible fashion and written in standard English?

Reviewer #1: Yes

Reviewer #2: Yes

5. Review Comments to the Author

Reviewer #1: I am confused.

It is well written paper, although some parts are unclear for readers.

If I understand it is a part of SOONER Trial but what was the aim of that paper?

Major

The simulation protocol is unclear, especially in part simulation scenario.

What was the learning objective in that simulation scenario?

In results we don't have any results of the scenario activity. We don't know in what percentage naloxone was administrated and if CPR was performed. If yes what was the activity timeline. When naloxone was administrated, when 911 call was performed? What kind of device/protocol was used for quality of CPR judgment? Chest compression fraction - CCF?

In fact we don know if CPR was necessary in scenario and if any of participants performed it before emergency arrival. So in fact the title of manuscript should be rephrased. Resuscitation is the first word in the title but we don't know if resuscitation was the learning objective or was performed in any case?(!) Or mayby I am wrong. if yes it should be indicated what is the aim of paper study - on fact the results include only debriefing part.

Minor

Please explain all abbreviations.

The participants demographics and table 1 should be included in methods part. The participants profile is not the result of the presented study!

Reviewer #2: Thank you for sending me this manuscript for review.

The objective of this research is to describe the simulation experience of non-clinical participants for cardiopulmonary resuscitation training as a treatment for cardiorespiratory arrest due to overdose.

The manuscript in general brings great novelties to scientific knowledge, however, I think that it requires some reflections that I comment on below:

I think that the qualitative design provides a lot of information, however, a mixed design with validated instruments useful in simulation would have helped to recruit more evidence to the scientific community. There are instruments such as Student Satisfaction and

Self-Confidence in Learning Scale” that assesses the level of satisfaction and self-confidence of the participants in a simulation activity or the “Debriefing experience sclae” that offers information about the debriefing experience. I suggest that in later studies a mixed design can be considered.

Regarding the simulation protocol used. I understand that the SCOONER study was standardized, but I think that using an actor instead of a mannequin would have helped the participants to experience the situation more realistically. However, I see from the results that this was not a limitation for them. Likewise, using simulation strategies by repetition, such as the rapid cycle, I think would have helped the participants to increase their learning curve regarding resuscitation skills in the event of cardiorespiratory arrest due to overdose.

To understand how the post-simulation reflections were carried out, it is important to describe the debriefing process and what methodology was used. I suggest that it can be added to the manuscript.

Regarding ethical considerations, it is not specified how the participants were informed, if they signed an informed consent and how said data was treated. I think it is important, due to the profile of the participant, to specify these data for the publication of this manuscript.

I consider your post, with some minor changes

6. PLOS authors have the option to publish the peer review history of their article (what does this mean?). If published, this will include your full peer review and any attached files.

Reviewer #1: No

Reviewer #2: No

---

## [Author Response · Author response to Decision Letter 0]

13 May 2022

Thank you for taking the time to review our study for consideration of publication. We would like to address the following journal requirements.

1. We have updated file naming for the table, figures and supporting documentation to reflect PLOS ONE's style templates.

2. We have changed the term "Caucasian" to "White" as recommended. Thank you for flagging this issue. We have also included additional supporting documentation to enhance the replicability of our study. We have included an outline of the briefing process, the debriefing guide and the interview guide and have appropriately referenced these supporting documents in the text.

3. In the case of this qualitative study, raw data refers to interview transcripts with study participants. During these interviews participants are asked about deeply personal and sensitive issues and individuals share information that could potentially lead to their identification. Due to ethical restrictions by the Unity Health Toronto Research Ethics Board, the individual participant-level data underlying this study may not be made publicly available. However, interested and qualified researchers who meet criteria for access to confidential data may submit data access requests to Unity Health Toronto at researchcontracts@smh.ca.

4. Our cover letter has been revised to address ethical restrictions to sharing the data set as outlined above. Supporting documentation has also been updated to include the simulation and study protocols to improve the transparency and replicability of our study.

5. The ethics statement has moved to the Methods section as recommended and we have revised it to include the full name of the IRB who approved the study and a more detailed description of how consent was obtained from participants.

---

## [Decision Letter · Decision Letter 1]

21 Jun 2022

Resuscitation simulation among people who are likely to witness opioid overdose:

Experiences from the SOONER Trial

PONE-D-21-30823R1

Dear Dr. Whittall,

We’re pleased to inform you that your manuscript has been judged scientifically suitable for publication and will be formally accepted for publication once it meets all outstanding technical requirements.

Kind regards,

César Leal-Costa, Ph. D

Academic Editor

PLOS ONE

Additional Editor Comments (optional):

Reviewers' comments:

Reviewer's Responses to Questions

**Comments to the Author**

1. If the authors have adequately addressed your comments raised in a previous round of review and you feel that this manuscript is now acceptable for publication, you may indicate that here to bypass the “Comments to the Author” section, enter your conflict of interest statement in the “Confidential to Editor” section, and submit your "Accept" recommendation.

Reviewer #1: All comments have been addressed

2. Is the manuscript technically sound, and do the data support the conclusions?

Reviewer #1: Yes

3. Has the statistical analysis been performed appropriately and rigorously? 

Reviewer #1: Yes

4. Have the authors made all data underlying the findings in their manuscript fully available?

Reviewer #1: Yes

5. Is the manuscript presented in an intelligible fashion and written in standard English?

Reviewer #1: Yes

6. Review Comments to the Author

Reviewer #1: Congratulation. I have no additional comments.

It is well prepared manuscript and sufficient for publication.

7. PLOS authors have the option to publish the peer review history of their article (what does this mean?). If published, this will include your full peer review and any attached files.

Reviewer #1: No

---

## [Editor Report · Acceptance letter]

24 Jun 2022

PONE-D-21-30823R1 

Resuscitation simulation among people who are likely to witness opioid overdose:
Experiences from the SOONER Trial 

Dear Dr. Whittall:

I'm pleased to inform you that your manuscript has been deemed suitable for publication in PLOS ONE. Congratulations! Your manuscript is now with our production department. 

Kind regards, 

on behalf of

Dr. César Leal-Costa 

Academic Editor

PLOS ONE